# Laparoscopic Peritoneal Wash Cytology-Derived Primary Human Mesothelial Cells for In Vitro Cell Culture and Simulation of Human Peritoneum

**DOI:** 10.3390/biomedicines9020176

**Published:** 2021-02-10

**Authors:** Myriam Holl, Lucas Becker, Anna-Lena Keller, Nora Feuerer, Julia Marzi, Daniel A. Carvajal Berrio, Peter Jakubowski, Felix Neis, Jan Pauluschke-Fröhlich, Sara Y. Brucker, Katja Schenke-Layland, Bernhard Krämer, Martin Weiss

**Affiliations:** 1Department of Women’s Health, Eberhard Karls University, 72076 Tübingen, Germany; myriam.holl@student.uni-tuebingen.de (M.H.); lucas.becker@uni-tuebingen.de (L.B.); nora.feuerer@nmi.de (N.F.); julia.marzi@uni-tuebingen.de (J.M.); Daniel.Carvajal-Berrio@med.uni-tuebingen.de (D.A.C.B.); peter.jakubowski@med.uni-tuebingen.de (P.J.); felix.neis@med.uni-tuebingen.de (F.N.); jan.pauluschke-froehlich@med.uni-tuebingen.de (J.P.-F.); sara.brucker@med.uni-tuebingen.de (S.Y.B.); kschenkelayland@me.com (K.S.-L.); bernhard.kraemer@med.uni-tuebingen.de (B.K.); 2NMI Natural and Medical Sciences Institute, University of Tübingen, 72770 Reutlingen, Germany; anna-lena.keller@nmi.de; 3Cluster of Excellence iFIT (EXC 2180) Image-Guided and Functionally Instructed Tumor Therapies, Eberhard Karls University, 72076 Tübingen, Germany; 4Department of Medicine/Cardiology, University of California Los Angeles (UCLA), Los Angeles, CA 90095, USA

**Keywords:** mesothelial cells, primary cell culture, in vitro cell culture, 2D/3D cell culture model, human peritoneum, laparoscopy, wash cytology, isolation methodology

## Abstract

Peritoneal mucosa of mesothelial cells line the abdominal cavity, surround intestinal organs and the female reproductive organs and are responsible for immunological integrity, organ functionality and regeneration. Peritoneal diseases range from inflammation, adhesions, endometriosis, and cancer. Efficient technologies to isolate and cultivate healthy patient-derived mesothelial cells with maximal purity enable the generation of capable 2D and 3D as well as in vivo-like microfluidic cell culture models to investigate pathomechanisms and treatment strategies. Here, we describe a new and easily reproducible technique for the isolation and culture of primary human mesothelial cells from laparoscopic peritoneal wash cytology. We established a protocol containing multiple washing and centrifugation steps, followed by cell culture at the highest purity and over multiple passages. Isolated peritoneal mesothelial cells were characterized in detail, utilizing brightfield and immunofluorescence microscopy, flow cytometry as well as Raman microspectroscopy and multivariate data analysis. Thereby, cytokeratin expression enabled specific discrimination from primary peritoneal human fibroblasts. Raman microspectroscopy and imaging were used to study morphology and biochemical properties of primary mesothelial cell culture compared to cryo-fixed and cryo-sectioned peritoneal tissue.

## 1. Introduction

Human peritoneal mesothelial cells form monolayers and face the mucosa of the peritoneal cavity. They grow on a thin basement membrane (peritoneum), connected with tissue stroma (collectively referred to as the serosa). Peritoneal mesothelial cells surround internal organs, form a barrier, and are the first line of defense. They ensure a protective, slippery and non-adhesive surface to facilitate the non-viscous movement of inner organs by the secretion of peritoneal fluid containing nutrients, phospholipids, surface glycosaminoglycans such as the polysaccharide hyaluronan [1]. The peritoneum is an important immunological interface, continuously recruiting leucocytes from the blood by cytokines and chemokines. Under physiological conditions, 50–90% of the peritoneal leucocyte population represents monocytes and macrophages, ensuring the defense of pathogens, clearance of apoptotic cells and regulating extracellular matrix (ECM) composition and sensing [2]. In light of peritoneal injury, mesothelial cells divide, transform into spindle-shaped cells and migrate to the wound area when activated. The mesothelial healing processes arise diffusely across the injured surface, in contrast to fibroblasts, which start to grow from wound edges [3].

If the physiological homeostasis fails, several peritoneal diseases may arise. Peritoneal injury and inflammation often form symptomatic and painful adhesions between serosal surfaces by edema and increased fibrogenic exudates [2,4,5,6]. Besides serious disorders of organ function, the formation of peritoneal adhesions is responsible for 15–20% of infertility cases after gynecologic surgery [7]. The development of intraperitoneal fibrosis during long-term peritoneal dialysis (PD) is a leading problem for the therapy of chronic kidney failure [8]. Neoplastic spreading along the peritoneum of either primary neoplasms from mesothelial cells (mesothelioma) or cancer cells from extraperitoneal (e.g., colorectal cancer and breast cancer) and intraperitoneal (e.g., ovarian cancer) lesions is a crucial factor for the oncological outcome [9]. Besides malignant cell growth, the peritoneum is frequently involved in the pathological appearance of benign ectopic cells such as endometriosis.

All of these diseases have a serious clinical impact; however, the underlying pathogenesis for each is, by far, not fully understood, and effective treatment procedures are mostly pending. Therefore, the characterization and analysis of peritoneal tissue cells have become an increasing research focus. The sufficient isolation and cultivation of primary human mesothelial cells from the peritoneum offer multiple possibilities to investigate physiology, pathogenesis and drug testing in conventional 2D in vitro models as well as in vivo like fluidic 3D platforms. Different isolation methods for human mesothelial cells were established over the past years. However, most of the widely used methods like cell isolation from omentum tissue samples face several disadvantages, such as heavy fibroblast (F) contamination, as well as limited proliferation ability and early senescence of cells [10,11]. Depending on the density of blood vessels within the tissue sample, contamination with microvascular endothelial cells and consecutive distortion of experimental results is likely [12,13]. Other working groups described the isolation of human mesothelial cells from irrigation fluid after PD. This simple method allowed fast and easily reproducible isolation of mesothelial cells; however, cells of chronic PD patients showed several degenerative changes (enlargement, multivacuolation and reduced function of cell organelles) due to the underlying disease and the long-time exposure to PD solution and its ingredients [14,15]. Here we report a new and easily reproducible isolation method of peritoneal primary human mesothelial cells from peritoneal wash cytology (PWC). The established method allows long-term cultivation with the highest purity of mesothelial cells. Cells were deeply characterized by immunofluorescence (IF-) staining, flow cytometry, Raman microspectroscopy and Raman imaging.

## 2. Materials and Methods

### 2.1. Isolation of PWC-Derived Primary Mesothelial Cells

PWC was obtained from consenting patients undergoing elective laparoscopic surgery at the Department of Women’s Health in Tübingen. To increase cell adherence, the culture flask was coated with 0.1% gelatin diluted in sterile water. T75 cell culture flasks (Greiner bio-one, Frickenhausen, Germany) were coated with 0.1% gelatin from bovine skin (Typ B powder, Sigma-Aldrich, Darmstadt, Germany) diluted in sterile water. Sterile phosphate-buffered saline (PBS) (Dulbecco’s phosphate-buffered saline, 14190-094, ThermoFisher, Schwerte, Germany) was used for all washing steps. An amount of 5 mL of 0.1% gelatin was added to a T75 cell culture flask. The flask was shaken gently until the whole bottom was covered with liquid. The flask was stored at 4 °C and incubated for 2 h. After incubation time, the supernatant was removed. The flask was washed with PBS two times. The coated flask can be stored at 4 °C for up to 5 days until usage.

PWC was taken under sterile conditions during otherwise indicated laparoscopic surgeries at the Department of Women’s Health in Tübingen between February and November 2020 after written informed consent of the patients. The Ethical Committee of the Medical Faculty of the Eberhard-Karls-University Tübingen approved the scientific use of the tissue of the Medical Faculty of the Eberhard-Karls-University Tübingen (649-2017BO2, approval: 12.01.2018 and 495/2018BO2, approval: 19.10.2018).

Isolated human peritoneal mesothelial cells were cultured in DMEM/F-12 + GlutaMAX (21331020, ThermoFisher, Schwerte, Germany) containing 10% Fetal Bovine Serum (FBS, 10270-106, ThermoFisher, Schwerte, Germany), 1% Penicillin/Streptomycin (15140-122, ThermoFisher, Schwerte, Germany), 1% L-Glutamine (25030-024, ThermoFisher, Schwerte, Germany). The cells were cultured at 37 °C and 5% CO_2_. Confluent cells were washed with PBS and incubated with 5 mL of Trypsin-EDTA (0.05%) (Trypsin- Ethylenediaminetetraacetic acid, 25300-054, ThermoFisher, Schwerte, Germany) at 37 °C and 5% CO_2_ until cells lose adherence. 7 mL of culture medium was added, transferred to a falcon tube and centrifuged at 300× *g* for 3 min. After removing the supernatant, cells were resuspended in 4 mL of fresh prewarmed culture medium and seeded again. Cell culture was observed by light microscopy (EVOS XL Core Cell Imaging System, ThermoFisher, Bothell, WA, USA).

### 2.2. Isolation of Primary Human Peritoneal Fibroblasts

Fibroblasts were isolated from peritoneal tissue samples obtained from consenting patients that were undergoing a cesarean. The Ethical Committee of the Medical Faculty of the Eberhard-Karls-University Tübingen approved the scientific use of the tissue of the Medical Faculty of the Eberhard-Karls-University Tübingen (649-2017BO2, approval: 12.01.2018). The tissue sample was washed two times with PBS. After removal of fatty tissue and blood vessels, the sample was prepared according to Takashima et al. [16]. Small slices were placed into a 6-well plate and covered with 1 to 1.5 mL of culture medium (MEM, Minimum Essential Media, 31095029, ThermoFisher, Schwerte, Germany), containing 10% FBS, 1% Penicillin/Streptomycin, 1% L-Glutamine. After 7 days, cells were detached with 0.05% Trypsin/EDTA and transferred to a T75 culture flask.

### 2.3. Primary Human Peritoneal Tissue Samples

Peritoneal tissue samples were obtained from consenting patients undergoing elective cesarean section at the Department of Women’s Health in Tübingen. The sample was stored and transported at 4 °C in DMEM/F-12 cell culture media for further processing within 6 h. The Ethical Committee of the Medical Faculty of the Eberhard-Karls-University Tübingen approved the scientific use of the tissue of the Medical Faculty of the Eber-hard-Karls-University Tübingen (495/2018BO2, approval: 19.10.2018)

### 2.4. HaCat (Primary Human Keratinocyte Cells) Culture, BJ Fibroblasts (Primary Human Fibroblasts)

HaCats (Primary human keratinocyte cells, CLS, 300493) and BJ fibroblasts (primary human fibroblasts, ATCC CRL-2522) from human skin were cultured in DMEM + GlutaMAX (31966021, ThermoFisher, Schwerte, Germany) containing 10% FBS, 1% Penicillin/Streptomycin and 1% L-Glutamine.

### 2.5. Immunofluorescence

Isolated mesothelial cells at passage 1–4 and isolated fibroblasts at passage 2–5 were used for immunofluorescence analysis. Three independent experiments were performed using cells of different patients. 1 × 10^5^ mesothelial cells were seeded at gelatin-coated dishes (µ-Dish 35 mm, low, ibidi) according to the given instructions. Fibroblasts were seeded with 5 x 10^4^ cells per dish. After 24 h, cells were washed and fixed in 4% Paraformaldehyde (PFA, P6148, Sigma-Aldrich, Darmstadt, Germany) for 10 min at 37 °C. Cells were permeabilized with 0.1% Triton-X 100 (3051.3, ROTH, Karlsruhe, Germany) for 15 min at room temperature, followed by blocking in 2% bovine serum albumin (BSA, 8076, Karlsruhe, Germany) in order to reduce nonspecific binding for 60 min. Afterward, the cells were incubated with the primary antibodies, listed in Table 1, diluted in 0.1% BSA overnight at 4 °C.

Subsequently, samples were incubated with the appropriate secondary antibodies (AlexaFluor 647 anti-rabbit IgG, AlexaFluor 488 anti-mouse IgG1 and AlexaFluor 488 anti-mouse IgG2b (1:1000 diluted in 0.1% BSA; Thermo Fisher Scientific)) for 45 min at room temperature, protected from light. Finally, nuclei were stained with Hoechst 33,342 (R37605, ThermoFisher, Schwerte, Germany) according to the given instructions, for 20 min in the dark. Samples were analyzed using a fluorescence microscope (Cell Observer, Carl Zeiss AG, Oberkochen, Germany).

### 2.6. Flow Cytometry

For flow cytometric analysis, cells were prepared in the following manner: Mesothelial cells (at passage one or two), fibroblasts (at passage four to six), BJ fibroblasts and HaCats were used. Cells were harvested with trypsin and EDTA (0.05%, 10 mM at 37 °C for 5 min) and centrifuged at 300× *g* for 3 min. 5 × 10^5^ cells were resuspended in 1 mL washing buffer (2% heat-inactivated FCS, 0.05 mM EDTA and 0.05% NaN_3_ in PBS, pH = 7.4) and centrifuged at 300× *g* for 5 min. Nonspecific binding sites were blocked using 10% human male serum (diluted in washing buffer) for 20 min at 20–23 °C. Cells were centrifuged again at 300× *g* for 5 min. The supernatant was removed. Each pellet was resuspended in 250 µL BD Cytofix/-perm (Fixation/Permeabilization Solution Kit, 554714, BD Bioscience, Heidelberg, Germany) and incubated for 20 min on ice in the dark. Cells were centrifuged at 300× *g* for 5 min at 4 °C. Pellets were washed two times with BD Perm/Wash (Fixation/Permeabilization Solution Kit, 554723, BD Bioscience, Heidelberg, Germany). The cytokeratin antibody (cytokeratin antibody, anti-human, FITC, REAfinity, clone REA831, 130-112-931, miltenyibiotec, Bergisch Gladbach, Germany) was diluted 1:50 in water containing 10% BD Perm/Wash and 10% human serum. Cells were incubated for 30 min on ice in the dark and afterward washed with BD Perm/Wash and resuspended in 100 µL washing buffer. Cells were immediately analyzed using a BD FACSFortessa instrument (BD Bioscience, Heidelberg, Germany) with FACS DIVA software v9 (BD Biosciences, Heidelberg, Germany). Data were analyzed using FlowJo software v10 (FlowJo LLC, Ashland, VA, USA). Forward- and side-scatter (FSC-H and SSC-H) characteristics were used to exclude dead cells. Forward scatter area and height (FSC-A and FSC-H) characteristics were used to exclude cell doublets (see Appendix A).

### 2.7. Raman Analysis

Mesothelial cells isolated from PWC (MC-cc) at passage one to two were seeded with 2 × 10^5^ cells per (gelatin-coated) dish (µ-Dish 35 mm, high glass bottom, 81158, ibidi, Gräfelfing, Germany). After two days of culture, cells were fixed in 4% PFA at 37 °C for 15 min prior to Raman measurements. Cryopreserved human tissues of the peritoneum (MC-cryo) were cut (10 µm thickness) and collected on glass coverslips. Prior to Raman imaging, tissue sections were washed with PBS for 10 min. Tissues and fixed cells were covered in PBS during the entire Raman measurement to prevent sample burning and dehydration. Spectral mapping was performed on a customized upright WITec alpha300 R Raman system (WITec GmbH, Ulm, Germany) equipped with a green laser (532 nm) and a CCD spectrograph with a grating of 600 g/mm. Raman images were acquired at three different positions for MC-cc (*n* = 3) and MC-cryo (*n* = 2) at a laser power of 57 mW, an integration time per spectrum of 0.05 s and a pixel resolution of 1 × 1 µm using a 63x dipping objective (N.A. 1.4; Olympus, Tokyo, Japan)).

Raman image analysis of the spectral maps was performed with the software Project Five version 5.2; Software for image editing and image processing (WITec GmbH, Ulm, Germany). Raman data were subjected to cosmic ray removal, polynomial baseline correction to remove the glass background, and intensity normalization (area to 1). To analyze Raman image maps, true component analysis (TCA) was used at the spectral range of 700–1800 cm^−1^. In brief, TCA allows for the identification of spectral components dominating the spectral map. As a non-negative matrix factorization-based multivariate data analysis tool, this method defines similar spectra occurring in the whole scan as the same component and identified all pixels that provided these spectral characteristics in a false-color intensity distribution heatmap. TCA allowed preselecting regions of interest (ROI) with similar spectral information representing nuclei, ECM, and lipids for both 2D cell culture and cryosections, which were extracted for further in-depth analysis of the molecular composition by principal component analysis (PCA) using Unscrambler X10.5 (Camo Software AS, Oslo, Norway). In brief, PCA is a classic multivariate data analysis tool for reducing the dimensionality of spectral data sets, increasing interpretability but minimizing information loss on a vector-based approach. By axis-rotation, a new set of axes, so called principal components (PCs) containing maximal directions of variance within a dataset are created. Plotting PC values against each other allow visualization of correlation and separation of two or more data sets. Underlying spectral differences can be interpreted based on the corresponding loading plot.

### 2.8. Image and Data Analysis

Statistical comparisons were performed from a minimum of three independent experiments. Data are given as mean ± SD (Standard Deviation). Statistical analysis was performed using GraphPad Prism v8 for Windows. Immunofluorescence images were prepared on ZEN v3.0 (blue edition, Oberkochen, Germany). Bright-field images were prepared using PowerPoint (v2019, Microsoft, Redmond, USA). FACS data were analyzed by FlowJo (software v10, LLC, Ashland, VA, USA).

## 3. Results

### 3.1. Isolation Methodology and Cell Culture of PWC-Derived Mesothelial Cells

Figure 1 summarizes the methodology that we have developed for the isolation of primary human mesothelial cells. It includes the obtaining of PWC during otherwise indicated gynecological laparoscopic surgeries of healthy women at the Department of Women’s Health of the University Hospital Tübingen, Germany, after written informed consent. The scientific use of human tissue samples was approved by the institutional review board of the medical faculty of the University Hospital Tübingen (ethical vote: 649-2017BO2, approval: 12.01.2018; and 495/2018BO2, approval: 19.10.2018). The patient-derived samples included eight PWC from healthy female patients with a median age of 35.8 years, average gravidity and parity of 1.5 and 0.5, respectively (Table 1). 6 (75%) out of 8 patients were premenopausal. One patient (12.5%) with absent menstrual bleeding > 6 months and < 2 months was defined as perimenopausal. Another patient (12.5%) was after benign hysterectomy with unknown ovarian function. Samples were taken immediately after intraabdominal access to avoid any contamination with erythrocytes from abdominal bleeding. All of the surgeries were performed due to benign reasons. After inspection of the anatomical area, the pouch of Douglas was repeatedly flushed with 10–20 mL of conventional sterile isotonic 0.9% sodium chloride (NaCl) solution for 30 s (Figure 1a). The liquid containing the detached mesothelial cells was collected with a standard syringe. The sample was stored and transported at 4 °C for further processing within 6 h. We recommend processing the samples as soon as possible after surgical collecting. Intraoperative findings included peritoneal adhesions in 5 patients (62.5%). One patient (12.5%) suffered from deep-infiltrating endometriosis. Primary human fibroblasts (F) (used as controls) were isolated from primary human peritoneal tissue samples obtained during Caesarean section of 3 patients with the mean age of 28.7 years (Table 2).

Following intraabdominal extraction, the PWC was further purified and centrifuged to remove contaminating cells and cell debris (Figure 1b). Therefore, the liquid was transferred to a sterile 50 mL falcon tube. Cells were spun at 300× *g* for 5 min. The supernatant was removed carefully, and 10 mL of fresh sterile PBS^-^ was added. The cell pellet was resuspended gently. After that, cells were spun again at 300× *g* for 5 min. Once again, the supernatant was removed. An amount of 4–5 mL of fresh prewarmed culture medium (DMEM/F-12 + GlutaMAX, containing 10% fetal calf serum, 1% Penicillin-Streptomycin and 1% L-Glutamine) was added. Optionally, cells were counted to determine cell number. The culture flask was coated with 0.1% gelatin diluted in sterile water to increase cell adherence. 7–8 mL of prewarmed culture medium was added to a T75 culture flask. The cell suspension was transferred to the culture flask. Cells were cultured at 37 °C and 5% CO_2_ for at least two days. After 2–3 days, cells were carefully washed with prewarmed PBS, to remove non-adherent and dead cells. Fresh culture medium was added every 2–3 days.

After 24 h of cultivation, cells showed a bi- to multipolar, elongated morphology (Figure 1c). On day three, cells appeared oval and circular (data not shown). Following this initial growth phase, cells adapted a typical cobblestone-type appearance. After four to five days, mesothelial cells revealed a homogeneous monolayer with the absence of fibroblast contamination. Depending on the seeded cell number (1 × 10^5^–1 × 10^7^ cells per T75 cell culture flask), cells reached confluence after 7 to 10 days (Figure 1d). Cells were successfully cultured over five passages. Beyond this point, the mesothelial cells showed an increase in cell size and number of cell nuclei, which indicate the development of senescent-like cells. Mechanical irritation or long-time incubation with trypsin/EDTA was often followed by a loss of mesothelial cell-morphology and consecutive transformation to a fibroblast-like appearance.

### 3.2. IF-Microscopic Characterization of PWC-Derived Mesothelial Cells and Suitable Molecular Markers to Distinguish from Peritoneal Fibroblasts

Next, we evaluated different cellular factors to specifically characterize and distinguish mesothelial cells from peritoneal fibroblasts, possibly contaminating the PWC-derived mesothelial cells. Therefore, primary mesothelial cells derived from peritoneal wash-cytology and tissue-derived peritoneal fibroblasts were analyzed by high-resolution IF-microscopy following IF-staining with specific antibodies against cytokeratin, fibronectin, calretinin and WT-1 (Figure 2a–h). Mesothelial cells showed a specific and high expression of the epithelial markers cytokeratin and calretinin. The IF-signal of the intermediate filament cytokeratin seemed to increase from perinuclear to the outer cell membrane. The calcium-binding protein calretinin was also found intracellularly with a lower perinuclear IF-signal and some cells with very low overall expression. Both factors, cytokeratin and calretinin, were negative in fibroblasts, whereas fibroblasts exhibited a high expression of the glycoprotein fibronectin. Most of the mesothelial cells showed no expression of fibronectin; however, it was sporadically detectable in all PWC-derived mesothelial cell populations (Figure 2c,d). Co-staining with cytokeratin and fibronectin of co-cultured mesothelial cells and fibroblasts showed a sufficient cell type-specific and spatial distinction of the two cell types suitable for high-resolution IF-microscopy (Supplemental Appendix A). In both cell types, the nuclear transcription factor WT-1 (Wilms’ tumor protein) was highly expressed within the cell nuclei. Overall, due to the highly specific, homogenous and intense expression in mesothelial cells irrespective of the passage number, cytokeratin especially was found as a suitable, reliable and simply performable marker for MC characterization and discrimination from other peritoneal cell types.

### 3.3. PWC-Derived Primary Mesothelial Cells Revealed above-Average Purity

By performing flow cytometry, the purity of PWC-derived mesothelial cell cultures was determined based on the expression of cytokeratin as a characterized mesothelial cell marker (see Figure 2). Following isolation from PWC and peritoneal tissue, the primary mesothelial cells and primary fibroblasts were harvested, formalin-fixed, and stained with specific antibodies against cytokeratin. After that, cells were analyzed by flow cytometry (Figure 3a–f,i). Flow cytometry was performed using a BD LSRFortessa cell analyzer with a 488 nm laser. Gates were set including 1% of unstained cells (for detailed gating strategy, please see Supplementary Appendix A). Results from one representative experiment showed an average mesothelial cell population of 98.9% being positive for cytokeratin (Figure 3a,c,e). Fibroblasts isolated from peritoneal tissue samples, instead, showed up to 16.1% of cells positive for cytokeratin (Figure 3b,d,f), which, by definition, cannot be fibroblasts. Figure 3i shows the result of three independent experiments with an average of 97.7 ± 0.7% of cytokeratin-positive cells within the PWC-derived mesothelial cells and 7.01 ± 5.5% of cytokeratin-positive cells within the peritoneal tissue-derived primary fibroblasts (Figure 3i). Additionally, we performed cytokeratin staining followed by flow cytometry of commercially purchased keratinocyte and BJ-fibroblast cell lines to compare their purity with our isolation method. Comparable to PWC, the keratinocyte cell line showed a proportion of 99.45 ± 0.6% cytokeratin-positive cells, whereas only 0.39% of cells of the BJ fibroblast cell line were positive for cytokeratin (Figure 3i, for detailed figures, please see Appendix A).

### 3.4. Characterization of the Molecular Structure and Composition of PWC-Derived Primary Mesothelial Cells Cryo-Preserved Peritoneal Tissue Samples Using Raman Imaging

To characterize PWC-derived primary mesothelial cells and native cryopreserved peritoneal tissue, we applied contactless and marker-independent Raman imaging and multivariate data analysis. Utilizing Raman imaging, we were able to perform morphological as well as molecular tissue analysis within the same experiment [17,18]. Raman spectra of primary mesothelial cell culture after PWC were acquired from three different patients after 2 days of cell culture in passage two. Additionally, we acquired Raman spectra of cryo-fixed and cryo-sectioned peritoneal tissue obtained from primary Caesarean sections of two patients at the Department of Women’s Health, Tübingen. Through Raman imaging and single spectra extraction, we were able to identify and compare the molecular characteristics and localization of different cellular components.

Raman imaging and True Component Analysis (TCA) allowed us to identify ECM-like intracellular spectra (green), ECM (orange), DNA within the nuclei (blue) as well as lipids (yellow) in both, primary mesothelial cell culture as well as cryo-fixed and cryo-sectioned peritoneal tissue (Figure 4a) based on their spectral fingerprints (Figure 4b).

Peaks related to DNA (788, 1095 cm^−1^; representing O-P-O stretching, see Table 3) [19,20] defined the structures detected by the first TCA component (Figure 4a, blue). Thereby, mesothelial cell culture and cryo-fixed peritoneal tissue were characterized by morphological differences in the size of nuclei (larger in mesothelial cell culture). Lipids (yellow) were detected as a second component of the TCA assigned to peaks at 1442, 1667, 1774 and 2852 cm^−1^ representing CH2 (methylene group) bending in fatty acids [21,22], unsaturated C=C bonds [21,22], carbonyl features [23] and CH2 symmetric stretch of lipids [24]. The lipid component is broadly distributed between nuclei in both analyzed tissues (Figure 4a, yellow). In mesothelial cell culture, a third component showed peak assignments for ECM-like spectra of proteins indicated by peaks at 929, 1005, 1567, and 2912 cm^−1^ representing C-C stretching in amino acids such as proline and valine [25], phenylalanine [19], Amide I [26] and CH stretch of proteins [24] (Figure 4a, green). In cryopreserved tissue sections, TCA identified different ECM protein components consisting of peaks related to collagen such as 818 [27], 1247 [27] and 1453 cm^−1^ [25] (Figure 4a, orange).

For in depth analysis of molecular differences between mesothelial cell culture and cryo-fixed peritoneal tissue, a PCA was performed on extracted single spectra of nuclei. The PCA scores plot revealed a distinct trend of clustering between both mesothelial cell conditions for all principal component (PC) except for PC3 showing a trend of spectral separation (Figure 4c). According to the correlating loadings plot (Figure 4e) trends of separation are explained by peaks at 918, 1269 and 1680 cm^−1^ contributing to mesothelial cell culture and 756, 1076, 1011, 1124, 1225, 1369, 1585 and 1628 cm^−1^ representing separation for cryo-fixed peritoneal tissue. The broad band at 918 cm^−1^ was assigned to proline [28], while 1269 cm^−1^ and 1680 cm^−1^ can be correlated to changes in Amide III and Amide I groups [29,30]. For peaks separating for cryopreserved tissue, higher tryptophan (Trp) content can be detected at 756 cm^−1^ [31]. Next to peaks representing C-C bond in carbohydrates (1011 cm^−1^ [22]) and lipids (1076 and 1124 cm^−1^ [20,28]), C-H bending vibrations are the reason for the trend of separation (1307 and 1369 cm^−1^ [28,31]). Additionally, Amide I peaks representing carbonyl groups at 1585 and 1628 cm^−1^ are separating the two groups slightly. T-tests of mean score values were performed to analyze the significance of clustering through all PCs (Figure 4g); however, they did not show any significant differences.

Similar results were found when comparing lipid features in mesothelial cell culture and cryo-fixed peritoneal tissue by PCA (Figure 4d) and t-tests on mean score values (Figure 4h). While PC1 (scores plot not shown) was rather linked to differences between tissue donors, PC2 indicated trends of separation due to peaks in the loading plot at 1296, 1445 and 1656 cm^−1^ contributing to mesothelial cell culture and peaks at 1009, 1102, 1240, 1384, 1589, 1628 and 1680 cm^−1^ for cryo-fixed peritoneal tissue samples (Figure 4f). Peak assignments within the loading plot are mainly assigned to differences in C-C vibrations (1296 cm^−1^ [30]) as well as CH2 wagging and CH2 scissoring (1302 and 1444 cm^−1^ [29,32]). (Figure 4f). Positive loadings of PC2 may be indicating a trend of separation for cryo-fixed peritoneal tissue revealed relatively higher Raman signals contributing from phenylalanine (1007 cm^−1^), carbohydrates (1002 cm^−1^) and Amide III (1233 cm^−1^). Additionally, peaks indicated more C=C vibrations at 1592 [33] and 1628 cm^−1^ [34] as well as C=O vibrations at 1683 cm^−1^. For lipid components, three principal components could be calculated and compared between the two mesothelial tissue specimens, which showed no statistically significant differences (Figure 4h).

## 4. Discussion

The human peritoneum fulfills multiple tasks to ensure the integrity of the human body. Since the peritoneum accounts for several benign and malignant diseases, the cultivation of patient-derived peritoneal tissue is important for the investigation of pathomechanisms and possible treatments. In this study, we established a new isolation method for human peritoneal mesothelial cells. In general, primarily isolated mesothelial cells from solid tissue samples are very likely to be contaminated with other cell types. Several early studies on the isolation of mesothelial cells from omental tissue samples by enzymatic degradation described a homogeneous population. Major concerns about the purity of these primary cell cultures were disproved by the examination of several cellular characteristics. In 1990, Stylianou et al. described their isolation method utilizing very low trypsin concentrations of 0.125% *wt*/*vol* to limit fibroblast contamination [11]. However, the influence of enzyme concentration and incubation time had a significant impact on the purity of isolated primary mesothelial cells, which challenges the method itself. To characterize the isolated mesothelial cells, they especially used IF-staining of cytokeratin and further likely specific intermediate filaments as well as electron microscopic demonstration of microvilli. Omental trypsin/ethylenediaminetetraacetic acid (EDTA) disaggregation is one of the most frequently used methods for mesothelial cells in literature [35,36,37]. Chung-Welch et al., in 1990, described a two-stage collagenase dissociation procedure to obtain mesothelial and endothelial cells from omental tissue [38]. Via brightfield microscopy, they show a mostly typical cobblestone-like appearance of cells after two days; however, some cells appear suspicious for fibroblasts due to a spindle-like, elongated phenotype. Interestingly, incubation with endothelial growth factor (EGF) increased the amount of fibroblast-like cells to approximately 50%.

Anatomically, mesothelial cells are connected with the basal lamina, a layer of collagen IV and laminin containing ECM, at the basal surface [9]. The submesothelial stroma consists of connective tissue consisting of collagen fibers, fibronectin, proteo- and glycosaminoglycans, (myo)fibroblasts, adipocytes as well as lymphatic tissue and (in a low density) microvessels. Generally, binding of the mesothelial cells to the basal lamina is weak and cells may have already detached due to minor injuries followed by exposed stroma. Therefore, we ensured to exclusively enroll patients with macroscopically preserved peritoneal integrity. Consequently, the collection of PWC was the very first step after entering the abdomen. Certain contamination with other stromal cell types is likely. However, a serious contamination would have been identified by an inferior ratio of cytokeratin positive and negative cells, because any of the stromal cell components (fibroblasts, lymphocytes, adipocytes and microvascular pericytes and endothelial cells) usually do express cytokeratin [10]. Due to the complete enzymatic digestion of the peritoneal tissue including the stroma, the contamination with microvascular endothelial cells is a major problem for mesothelial cell isolation from solid omental tissue [12]. Nevertheless, even in case of minor injuries of the peritoneal surface, microvascular endothelial cell contamination is highly unlikely for PWC due to shielded anatomical integrity of microcapillaries by fibroblastlike pericytes, longitudinally surrounding the microvessels and, thereby, preventing the direct contact of endothelial cells with PWC fluid [39].

Besides a considerable risk for contamination with other cell types (e.g., fibroblasts, endothelial cells), the use of solid visceral (e.g., omentum) or parietal (e.g., pelvic wall) peritoneal tissue is associated with ethical, methodological, and general scientific disadvantages. The excision of peritoneal tissue from healthy volunteers/patients is always critical due to the risk of injury, bleeding, and the damage of physiological structures. In the present study, we obtained peritoneal samples from the cutting edge during primary cesarean sections, in which the peritoneum is dissected, anyway. However, in order to ethically justify the tissue removal and the associated risks, it is usually carried out in the context of other medical indications—such as cancer surgery, or, as in our example, in the context of surgical delivery. Underlying diseases (such as cancer) or conditions of changed physiology (e.g., pregnancy) may have a critical influence on the following experiments after tissue removal [40]. Here, we established an isolation method for peritoneal mesothelial cells from laparoscopic PWC. Primarily, the method aims to enable the easy, time and cost sparing isolation of healthy mesothelial cells to generate physiological 2D and 3D cell culture models. Therefore, it is especially sufficient for human subjects with preserved peritoneal integrity. Healthy patients undergoing diagnostic laparoscopy for reasons of intra-abdominal pain or fertility diagnostics were enrolled. Our methodology is a further development based on the mesothelial cell isolation from irrigation fluid after PD, which was established by several working groups [14,15,41]. Besides the simplicity and high reproducibility of this method, cells of chronic PD patients showed several degenerative changes (enlargement, multivacuolation and reduced function of cell organelles) triggered by the underlying disease and the long-time exposure to PD solution and its ingredients, which could critically interfere with following experiments. Due to the often diagnostic character of laparoscopies, severe pathologies can be excluded even before the non-invasive sample collection by PWC.

In the present study, we confined our characterization and flow cytometric analysis to cytokeratin, which demonstrated to be highly specific for the distinction between the epithelial mesothelial cells and mesenchymal fibroblasts as other working groups have proven before [9,42,43,44]. The efficiency and reliability of cytokeratin staining were far superior to other intracellular and apparently specific molecules (Figure 2). Identifying mesothelial cells by cytokeratin and calretinin expression was previously shown by Yang et al. [45]. Indeed, also in our experiments, calretinin was specific for mesothelial cells and could be identified irrespective of passage number. However, the expression of the calcium-binding protein was found to be very inconstant and partly inexistent (Figure 2e,f). Fibronectin, however, was often described as a sufficient marker for the characterization of ECM producing mesenchymal cells, especially fibroblasts [11,41,46,47]. As other working groups before, we showed that mesothelial cells, which are known to produce ECM to a certain extent, were partly positive for fibronectin, too [11,48]. Data from flow cytometry showed that mesothelial cells isolated from PWC exhibited a very high mean purity of 97.67 ± 0.67% (Figure 3). Compared to the abundance of studies, characterizing their isolates by microscopic procedures, flow cytometry as high-throughput analysis of thousands of single cells has considerable advantages in this context and verified a nearly pure mesothelial cell population (Figure 3). The isolated cells showed a bipolar and elongated morphology during the initial growth phase (Figure 1). After three days of culture, cells adapted an oval and circular appearance and finally revealed a cobblestone-type structure when reaching confluency after 7–10 days. Moreover, cell morphology as well as the expression of the specific markers, was stable over at least four to five passages. Other working groups described a similar morphological development over time [11,49,50]. Mesothelial cells from omental tissue isolated by Styliano et al. reached confluency after 18 ± 1 days and were successfully cultured up to six passages [11]. Mesothelial cells isolated from spent dialysis fluid of PD patients developed a heterogeneous population. Cells need a long time to reach confluency [41,47]. Presumably, mesothelial cells were pre-exposed to systemic influences and dialysate, which results in possibly damaged cells.

Raman microspectroscopy and multivariate data analysis have been used for the contact and marker independent analysis and biochemical characterization of various human cell types and tissues [17,18,51,52]. Here, we characterized and compared the morphology and biochemical properties of 2D cell culture of isolated mesothelial cells from PWC and cryofixed solid peritoneal tissue (Figure 4). Compared to previous studies mostly using brightfield and IF-microscopy, the multiparametric Raman imaging is of great advantage to simultaneously determine morphological and functional characteristics in living and cryofixed tissue. Previously, Parlatan et al. in 2019 evaluated Raman spectroscopy as a non-invasive diagnostic tool for peritoneal endometriosis [53]. In 2016, Gaifulina et al. characterized the mesothelial cell layer of visceral peritoneum mainly by defining connective tissue (collagen) and nuclei [54]. To our best knowledge, this is the first study deeply characterizing biochemical and morphological aspects of human peritoneal tissue by Raman microspectroscopy and imaging. The assessment of in vitro cultivated mesothelial cells and the mesothelial layer of cryofixed peritoneum in the present study showed a size difference of nuclei in favor of isolated and in vitro cultivated mesothelial cells. Moreover, we found morphological differences in cell and tissue architecture. Whereas ECM by definition occurs exclusively extracellularly ECM-like spectral components were located in the perinuclear (e.g., as location of collagen synthesis) as well as in intercellular compartments of the in vitro cultured mesothelial cells ECM spectra in mesothelial cryosections (especially assigned to collagen, e.g., 818, 1453 cm^−1^) seemed to be for the most located in the extracellular space. For both samples, lipids (e.g., 1774, 2852, 1369, 1076 cm^−1^) were found intracellular and perinuclear. Overall, using PCA, the biochemical information of in vitro cultured mesothelial cells and mesothelial cryosections tended to differ (Figure 4c,d); however, the identified differences in DNA/nuclei (e.g., 1095, 788 cm^−1^) and in lipid components (differences in CH2 and C=C molecular vibrations) were not statistically significant for any of the evaluated PCs (Figure 4g,h). Remarkably, and in comparison with the mesothelial layer of cryofixed peritoneum, the molecular Raman signatures of nuclei and lipids of in vitro cultured mesothelial cells were highly consistent, represented by a nominal standard division (SD), even though the Raman data were obtained from three independent patients.

It is widely recognized that 3D microenvironments alter gene expression, cell plasticity and differentiation of different tissues, i.e., physical cues among others [55]. Isolated primary human peritoneal mesothelial cells from PWC enable further studies on cellular microenvironment, mechanotransduction and thereby induced cellular regulation processes as mechanical forces are crucial factors of peritoneal functionality in light of permanent organ movement and peristalsis.

In conclusion, we demonstrate an innovative and efficient method to isolate primary human peritoneal mesothelial cells from PWC. This outstanding purity and morphological and molecular stability over several passages is a great advantage to use the patient-specific cells in standard in vitro as well in advanced 3D and microfluidic, in vivo-like cell culture models to improve the understanding of the development of peritoneal diseases.

## 5. Conclusions

Isolation of primary human mesothelial cells from laparoscopic PWC enables the ethically sound generation of capable, patient-derived 2D cell cultures without increased invasiveness. Due to the high purity of the mesothelial cell population, the established methodology could significantly simplify the development of next generation in vitro models and increase the validity and reproducibility of research on peritoneal diseases and treatment strategies.

## Figures and Tables

**Figure 1 biomedicines-09-00176-f001:**
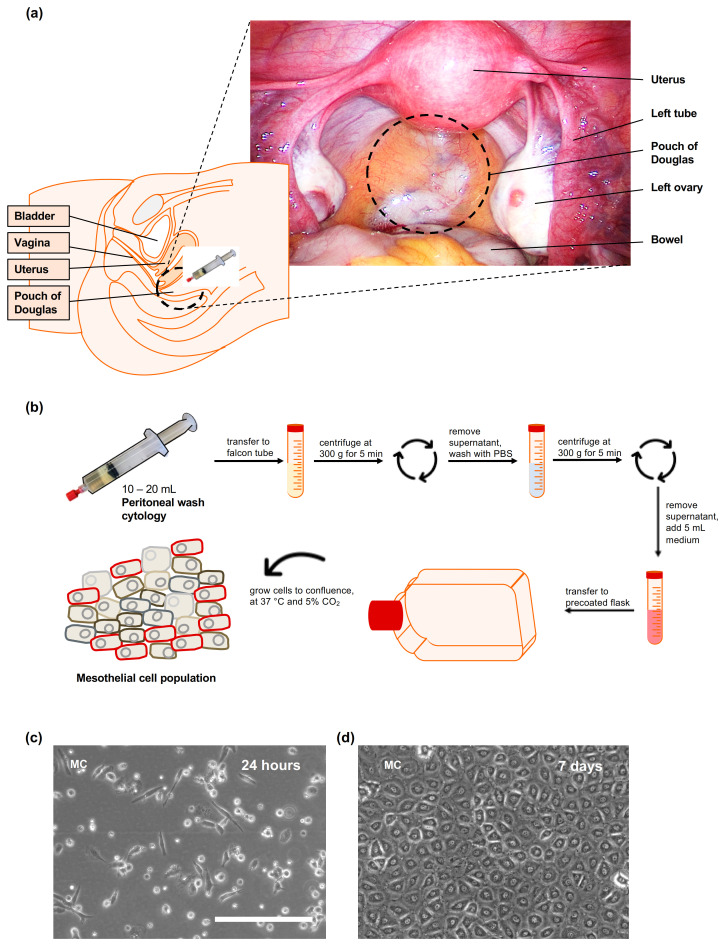
Isolation methodology of PWC-derived mesothelial cells. (**a**) Intraoperative laparoscopic view and schematic illustration of the pelvic anatomy. (**b**) Overview of the protocol for the isolation and cultivation of primary human peritoneal wash cytology-derived mesothelial cells prior to additional analysis. (**c**,**d**) Representative brightfield images of cultured primary human PWC-derived mesothelial cells after 24 h (**c**) and 7 days (**d**). Scale bar represents 200 µm for both images.

**Figure 2 biomedicines-09-00176-f002:**
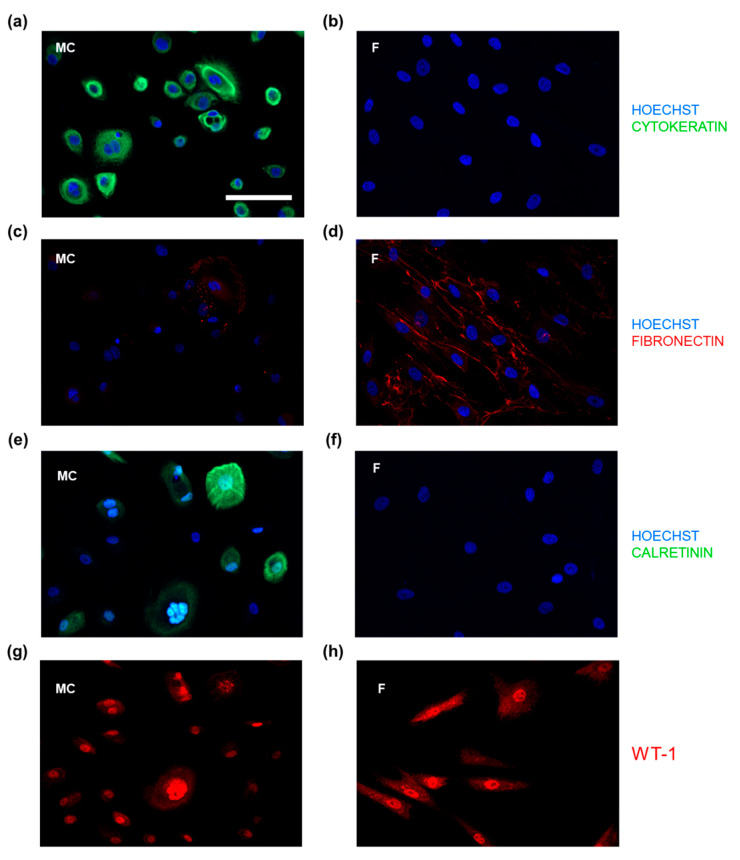
Characterization of PWC-derived mesothelial cells and distinction from peritoneal fibroblasts. Mesothelial cells (MC) were isolated from PWC, fibroblasts (F) were isolated from peritoneal tissue samples. Both cell types were cultured for 3 to 4 days. Immunofluorescence staining of MC and F for cytokeratin (**a**,**b**), fibronectin (**c**,**d**), calretinin (**e**,**f**) and WT-1 (Wilms’ tumor protein) (**g**,**h**) was performed. (**a**) MC showed a high expression of epithelial marker cytokeratin, while (**b**) F were found to be negative. (**c**) Some of MC showed a low expression of fibronectin, whereas (**d**) F exhibited a high expression of fibronectin. (**e**) MC expressed calretinin in varying intensities in contrast to (**f**) F, which were negative for calretinin. (**g**) MC and F (**h**) showed a high and similar expression of Wilms’ tumor protein 1 (WT-1). Scale bar represents 100 µm.

**Figure 3 biomedicines-09-00176-f003:**
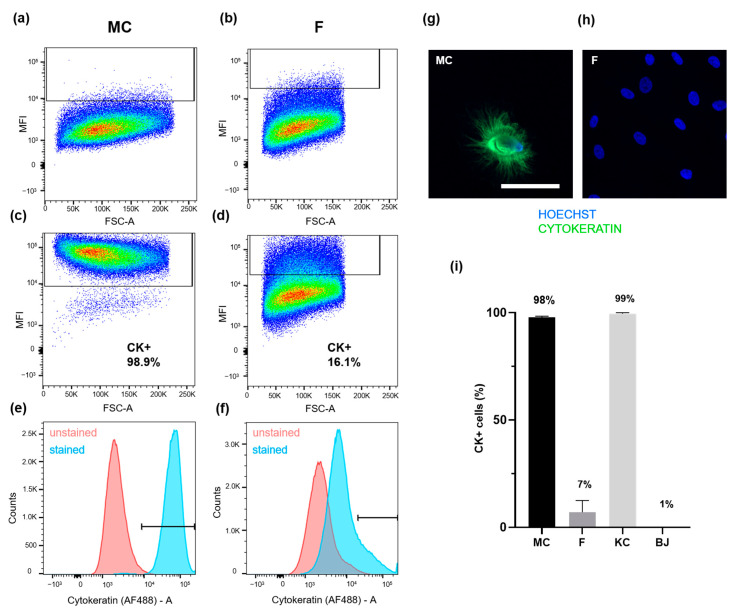
Determination of mesothelial cell (MC) purity by cytokeratin (CK) staining and flow cytometry. PWC-derived primary MC and peritoneal tissue-derived F, a commercially purchased keratinocyte (KC) and BJ fibroblast (BJ) cell line were stained for cytokeratin. Flow cytometry using the 488 nm laser of a BD LSRFortessa cell analyzer was performed to identify CK-positive (CK+) cells. (**a**,**b**) Mean fluorescence intensity (MFI) of unstained MC (**a**) and F (**b**) are shown. Both gates were set, including 1% of unstained cells. (**c**,**d**) Representative flow cytometry experiment. CK-positive cells are shown as FSC-A versus MFI. (**c**) 98.9% of the stained representative MC population was found to be cytokeratin-positive. (**d**) 16.1% of the representative F population expressed cytokeratin. (**e**,**f**) The histograms of unstained (red) and stained (blue) cells were compared. (**e**) CK-stained MC cells shifted substantially to the right, into the CK-positive gate. (**f**) CK-stained F population showed a small shift. (**g**,**h**) Representative IF-microscopy after CK-staining of MC (**g**) and F (**h**) in 63× magnification. (**i**) Mean percentage of CK-positive cells analyzed by flow cytometry and determined in three independent experiments. Results are shown as ± standard error of mean percentage. Scale bar represents 50 µm.

**Figure 4 biomedicines-09-00176-f004:**
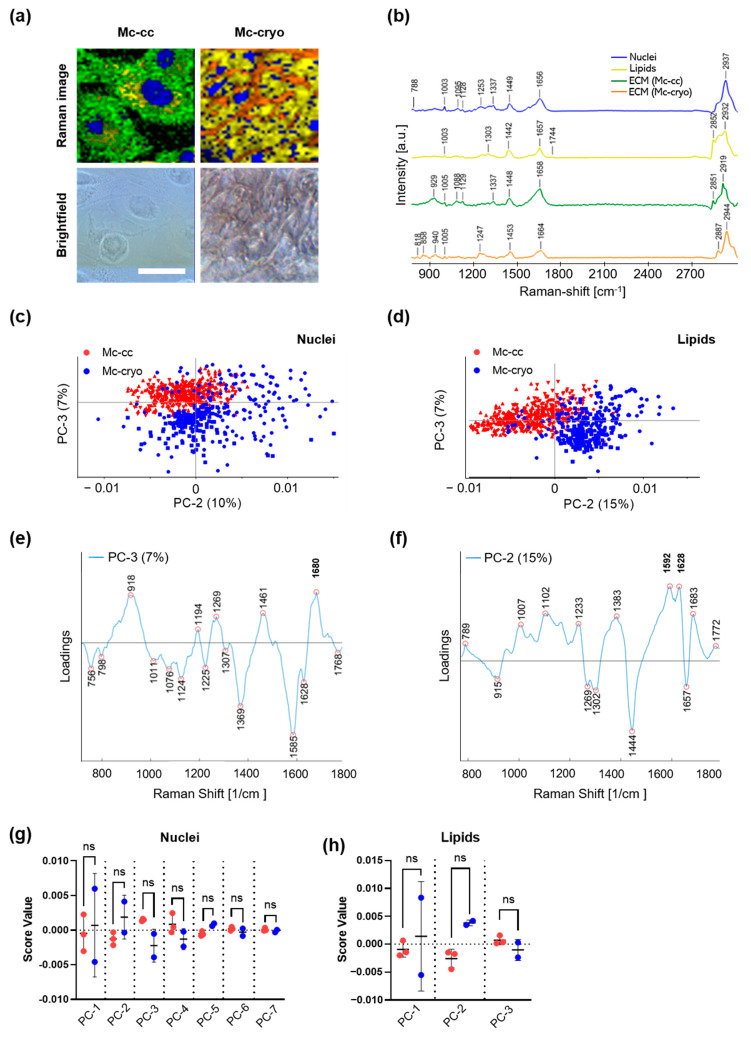
Raman imaging and multivariate data analysis of PWC-derived mesothelial cell culture (MC-cc) and cryo-fixed peritoneal tissue (MC-cryo). MC-cc were cultured on dishes, fixed with PFA and imaged in PBS using a customized upright WITec alpha300 R Raman system. MC-cryo were washed and imaged. MC-cc and MC-cryo were compared by multivariate data analysis with a focus on nuclei (DNA), lipids, and ECM components. (**a**) Representative Raman images and brightfield images of MC-cc and MC-cryo. Scale bar represents 50 µm. (**b**) Characteristic fingerprint spectra of the analyzed molecular components, utilized to identify nuclei/DNA (blue), lipids (yellow) and ECM (MC-cc, green; MC-cryo, orange) components by TCA. (**c**) Principal component analysis (PCA) scores plot of PC2 vs. PC3 of extracted nuclei spectra showed trends of clustering between MC-cc and MC-cryo. (**d**) Scores plot of PC2 vs. PC3 of extracted lipid spectra indicates a trend of clustering between MC-cc and MC-cryo. (**e**) Loadings plot corresponding to nuclei-specific PCA indicates peaks that explain the trend of clustering in PC-3. (**f**) Loadings plot of lipid-specific spectral information highlights peaks responsible for the trend of clustering in PC2. (**g**) DNA-specific average score values for PC1 to PC7 showed no significant difference between MC-cc and MC-cryo. (**h**) Lipid-specific average score values for PC1-PC3 revealed no significant difference between MC-cc and MC-cryo. Results are shown as mean ± SD of PC score values, *n* = 3 (MC-cc) *n* = 2 (MC-cryo).

**Table 1 biomedicines-09-00176-t001:** Mesothelial cell characterization.

Primary Antibody	IgG Source	Dilution	Application	Company
Cytokeratin–broad spectrum	Mouse (IgG1)	1:100	IF	Zytomed, Berlin, Germany
Fibronectin	Rabbit	1:100	IF	Abcam, Berlin, Germany
Calretinin	Mouse (IgG2b)	1:100	IF	Santa Cruz, Dallas, USA
Wilms’ tumor protein	Rabbit	1:100	IF	Santa Cruz, Dallas, USA

**Table 2 biomedicines-09-00176-t002:** Patient characteristics.

	MC	F
**No. of patients, *n* (%)**	**8 (100)**	**3 (100)**
Median age, years (range)	35.8 (22–55)	28.7 (25–31)
Gravidity, *n* (range)	1.5 (0–5)	1.0 (1)
Parity, *n* (range)	0.5 (0–2)	1.0 (1)
Ovarian function, *n* (%)		
premenopausal	6 (75.0)	3 (100)
perimenopausal	1 (12.5)	-
Cause for surgery		
Hypermenorrhea/Hysterectomy	2 (25.0)	-
Diagnostic/Pain	2 (25.0)	-
Diagnostic/Childwish	1 (12.5)	-
Cysts	3 (37.5)	-
Cesarean section	-	3 (100)
Intraoperative findings		
Adhesions	5 (62.5)	-
Endometriosis	1 (12.5)	-
none	2 (25.0)	-

**Table 3 biomedicines-09-00176-t003:** Identified Raman peaks [cm^−1^] and their molecular assignments.

Peaks (cm^−1^)	Assignment	Reference
756	Tryptophan	[31]
788	O-P-O stretching	[19]
818	C-C stretching in collagen	[27]
918	Proline	[28]
929	C-C stretching in amino acids such as proline and valine	[25]
1005	Phenylalanine	[19]
1011	Carbohydrates	[22]
1076	C-C in lipids	[20]
1095	DNA backbone	[20]
1124	Lipid backbone	[28]
1247	Amide III in collagen	[27]
1269	C-C vibration	[30]
1302	CH2 wagging	[32]
1369	Lipids	[31]
1442	C-C in fatty acids	[22]
1444	CH2 scissoring	[29]
1453	CH3 bending and CH2 scissoring in collagen	[25]
1592	C=C	[33]
1628	C=C	[34]
1657	C=C stretching	[21,22]
1658	Amide I	[26]
1680	Amide I	[29]
1774	Carbonyl feature in fatty acids	[23]
2852	CH2 symmetric stretch of lipids	[24]
2912	CH stretches of proteins	[24]

## Data Availability

The data presented in this study are available on request from the corresponding author. The data are not publicly available due to the ongoing character of the entire research project.

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
