# Peer review of "Laparoscopic Peritoneal Wash Cytology-Derived Primary Human Mesothelial Cells for In Vitro Cell Culture and Simulation of Human Peritoneum"

_biomedicines, 2021, doi:10.3390/biomedicines9020176_

Round 1

Reviewer 1 Report

In this paper authors present technique for the isolation and culture of primary human mesothelial cells from laparoscopic peritoneal wash cytology. The authors established a protocol 21 containing multiple washing and centrifugation steps, followed by cell culture at the highest purity and over multiple passages. Generally, studies addressing isolation, characterization and manipulation with primary cell cultures are of a great interest and represent a basement for future more reliable pharmacological models. Thus, it is of great importance to develop new methodologies and techniques in this field.

It was a pleasure to read nicely structured and well-prepared manuscript. Manuscript is written in clear, scientific sound manner. The topic is insightful and may attract great attention in future research. Moreover, the paper is well written and organized. The manuscript is well-structured and sound scientifically. Therefore, I recommend this paper for publication after minor revisions.

I understand that it is not a major topic of your study, but would be insightful for broad audience to add more discussion about 3D culturing of isolated mesothelial cells in a light of mechanical forces exerted by cellular microenvironment (Pharmaceuticals 2020, 13(12), 430; https://doi.org/10.3390/ph13120430).

Author Response

Authors’ response: We thank reviewer #1 for his positive statement and suggestions. We agree with the reviewer that it is insightful to discuss “3D culturing of isolated mesothelial cells in a light of mechanical forces exerted by cellular microenvironment”. Thus, we have added the following paragraph in the discussion part and cited the suggested literature:

“It is widely recognized that 3D microenvironments alter gene expression, cell plasticity and differentiation of different tissues, i.e. physical cues among others [1]. Isolated primary human peritoneal mesothelial cells from PWC enable further studies on cellular microenvironment, mechanotransduction and thereby induced cellular regulation processes as mechanical forces are crucial factors of peritoneal functionality in light of permanent organ movement and peristalsis.”

  1. Frtús, A., et al., Hepatic Tumor Cell Morphology Plasticity under Physical Constraints in 3D Cultures Driven by YAP–mTOR Axis. Pharmaceuticals, 2020. 13(12): p. 430.

Reviewer 2 Report

26-Jan-2021

Comments to the Authors

     I have reviewed the manuscript entitled “Laparoscopic Peritoneal Wash Cytology-Derived Primary Human Mesothelial Cells for In vitro Cell Culture and Simulation of Human Peritoneum” which has been submitted by Holl and coworkers. These researchers established a technology for the isolation and culture of primary human mesothelial cells from laparoscopic peritoneal wash cytology. They deeply characterize the mesothelial cells by immunofluorescence (IF) staining, flow cytometry, and RAMAN imaging spectroscopy to ensure high purity of cultured cells.

     This study was well designed and meticulously conducted, and acquired a new and easily reproducible method for cultivation of primary human peritoneal mesothelial cells. I believe the manuscript would be improved if the authors would address the following issues:

Major points:

  1. This study demonstrated that the expression of cytokeratin enabled obvious discrimination between the cultured cells from peritoneal fibroblasts. However, the expression of the more specific mesothelial cell marker “calrectinin” was inconsistent. Therefore, the authors should address how to exclude the contamination with other cytokeratin-positive cells, especially microvascular endothelial cells.

  2. This method may only be applied to human subjects with normal peritoneum, which may limit its use in exploration of various peritoneal diseases.

Author Response

Authors’ response, Comment 1 and 2: We also thank reviewer #2 for his positive statement and suggestions. We agree that it is beneficial to further address and discuss how to exclude the contamination with other cytokeratin-positive cells, especially microvascular endothelial cells (which is a major problem for cell isolation from solid tissue), which we discuss in detail in the following paragraph.
We further agree that the method is especially sufficient for human subjects with preserved peritoneal integrity. Primarily, the aim of the PWC isolation methodology is to enable the time and cost sparing easy isolation of healthy mesothelial cells to generate physiological 2D and 3D cell culture models. Therefore, we ensured to enrole patients with macroscopically preserved peritoneal integrity. The collection of PWC was the very first step after entering the abdomen.

To highlight the statements of reviewer #2 and according to his comments we have included the following two paragraphs into the discussion:

“Anatomically, mesothelial cells are connected with the basal lamina, a layer of collagen IV and laminin containing ECM, at the basal surface [2]. The submesothelial stroma consists of connective tissue consisting of collagen fibers, fibronectin, proteo- and glycosaminoglycans, (myo)fibroblasts, adipocytes as well as lymphatic tissue and (in a low density) microvessels. Generally, binding of the mesothelial cells to the basal lamina is weak and cells may already detached due to minor injuries followed by exposed stroma. Therefore, we ensured to exclusively enroll patients with macroscopically preserved peritoneal integrity. Consequently, the collection of PWC was the very first step after entering the abdomen. Contamination with other stromal cell types is unlikely because any of the stromal cell components (fibroblasts, lymphocytes, adipocytes and microvascular pericytes and endothelial cells) usually do express cytokeratin [3]. Due to the complete enzymatic digestion of the peritoneal tissue including the stroma, the contamination with microvascular endothelial cells is a major problem for mesothelial cell isolation from solid omental tissue [4]. Furthermore, even in case of minor injuries of the peritoneal surface, microvascular endothelial cell contamination is highly unlikely for PWC due to shielded anatomical integrity of microcapillaries by fibroblastlike pericytes, longitudinally surrounding the microvessels and thereby preventing the direct contact of endothelial cells with PWC fluid [5].”

“Primarily, the method aims to enable the easy, time and cost sparing isolation of healthy mesothelial cells to generate physiological 2D and 3D cell culture models. Therefore, is especially sufficient for human subjects with preserved peritoneal integrity.”

  1. Van Baal, J.O., et al., The histophysiology and pathophysiology of the peritoneum. Tissue Cell, 2017. 49(1): p. 95-105.
  2. Van Hinsbergh, V., et al., Characterization and fibrinolytic properties of human omental tissue mesothelial cells. Comparison with endothelial cells. Blood, 1990. 75(7): p. 1490-1497.
  3. Yung, S., F.K. Li, and T.M. Chan, Peritoneal Mesothelial Cell Culture and Biology. Peritoneal Dialysis International, 2006. 26(2): p. 162-193.
  4. Rayner, S.G. and Y. Zheng, Engineered Microvessels for the Study of Human Disease. Journal of biomechanical engineering, 2016. 138(11): p. 1108011-11080111.

Round 2

Reviewer 2 Report

  1. The added paragraph "Contamination with other stromal cell types is unlikely ...." should be revised as "likely".

  2. The add paragraph "Furthermore, even in case of minor injuries of the peritoneal surface, ....." should better be revised as "Nevertheless, even in case of minor injuries of the peritoneal surface, ....".

Author Response

Response to the reviewer

Reviewer # 2, Comment 1: The added paragraph "Contamination with other stromal cell types is unlikely ...." should be revised as "likely".

Authors’ response: We thank reviewer #1 for his suggestion. We agree with the reviewer that it is beneficial to lay the focus on a potential contamination with other cell types at that point. A simple change of unlikely into likely, however, would significantly change the literal meaning of the sentence. Thus, we have reworded the sentence concerned:

“Certain contamination with other stromal cell types is likely. However, a serious contamination would have been identified by an inferior ratio of cytokeratin positive and negative cells, [...]”

Reviewer #2, Comment 2: The add paragraph "Furthermore, even in case of minor injuries of the peritoneal surface, ....." should better be revised as "Nevertheless, even in case of minor injuries of the peritoneal surface, ....".:

Authors’ response, Comment 1 and 2: We thank reviewer #1 for his suggestion. We agree and changed the concerned “Furthermore” into “Nevertheless”.
